# Small RNA Targets: Advances in Prediction Tools and High-Throughput Profiling

**DOI:** 10.3390/biology11121798

**Published:** 2022-12-11

**Authors:** Katarína Grešová, Panagiotis Alexiou, Ilektra-Chara Giassa

**Affiliations:** 1Central European Institute of Technology (CEITEC), Masaryk University, 62500 Brno, Czech Republic; 2National Centre of Biomolecular Research (NCBR), Faculty of Science, Masaryk University, 62500 Brno, Czech Republic

**Keywords:** miRNA target prediction, small RNA target prediction, computational biology, machine learning, high-throughput sequencing

## Abstract

**Simple Summary:**

MicroRNAs (miRNAs) are a category of small RNAs (sRNAs) that have been found to regulate gene expression. Through the mediation of proteins from the Argonaute family, miRNAs target messenger RNAs (mRNAs) for destruction (cleavage or repression). Other types of sRNAs, including transfer-RNA-derived fragments (tRFs) and small interfering RNAs (siRNAs), have been indicated as potential regulators of gene expression. The complex network of RNA–RNA interactions is still under exploration, which can be assisted by the development of computational techniques. Here, we report the recent advancements in the field of bioinformatical and Machine Learning tools for the prediction of sRNA targets, and a brief overview of the development of high-throughput sequencing technologies.

**Abstract:**

MicroRNAs (miRNAs) are an abundant class of small non-coding RNAs that regulate gene expression at the post-transcriptional level. They are suggested to be involved in most biological processes of the cell primarily by targeting messenger RNAs (mRNAs) for cleavage or translational repression. Their binding to their target sites is mediated by the Argonaute (AGO) family of proteins. Thus, miRNA target prediction is pivotal for research and clinical applications. Moreover, transfer-RNA-derived fragments (tRFs) and other types of small RNAs have been found to be potent regulators of Ago-mediated gene expression. Their role in mRNA regulation is still to be fully elucidated, and advancements in the computational prediction of their targets are in their infancy. To shed light on these complex RNA–RNA interactions, the availability of good quality high-throughput data and reliable computational methods is of utmost importance. Even though the arsenal of computational approaches in the field has been enriched in the last decade, there is still a degree of discrepancy between the results they yield. This review offers an overview of the relevant advancements in the field of bioinformatics and machine learning and summarizes the key strategies utilized for small RNA target prediction. Furthermore, we report the recent development of high-throughput sequencing technologies, and explore the role of non-miRNA AGO driver sequences.

## 1. Introduction

RNA-induced gene silencing, also known as RNA interference (RNAi), is a widespread, evolutionary conserved mechanism. First described in Caenorhabditis elegans [1], it takes place when double-stranded RNA (dsRNA) molecules bring about the cleavage of an mRNA molecule with which they are at least partially complementary. RNAi is essentially triggered by small RNA fragments derived from long dsRNAs. Small RNAs (sRNA), first identified in Escherichia coli in 1984 [2], act to down-regulate the expression of target genes by the means of decreased translation and/or increased mRNA turnover [3]. They are universally found in all three kingdoms of life: Archaea, Bacteria, and Eukaryotes, where they adopt distinct mechanisms of RNAi. In eukaryotes, the most well-studied sRNAs are microRNAs (miRNAs, 18–26 nt) [4,5], small interfering RNAs (siRNAs, 20–27 nt) [6], and piwi-associated RNAs (piRNAs, 21–35 nt) [7]. miRNAs or siRNAs can assemble into a ribonucleoprotein complex called the RNA-induced silencing complex (RISC), composed of proteins such as RNA helicases, nucleases, and RNA-binding proteins [8]. piRNAs are found in animal germlines and their biogenesis from single-stranded RNA precursors involves primary processing by a set of proteins and the ping-pong cycle for amplification [9]. miRNAs, siRNAs, and piRNAs carry out their main functions (post-transcriptional mRNA cleavage, translational repression or decay, and transcriptional silencing) primarily by means of base-pairing with their DNA or RNA target. In eukaryotes, these RNA-silencing functions are mediated by the Argonaute family proteins, with the AGO sub-clade being associated with miRNAs and siRNAs, and the PIWI sub-clade with piRNAs [10]. For miRNAs, such interactions with their targets are classified as “canonical” when they are mediated by the “seed”, a region of 6–8 nt on the 5′ end of the sRNA that forms canonical (Watson–Crick) base pairs with the target [11,12].

In addition to their fully complementary on-targets binding, siRNAs and miRNAs can bind and regulate numerous miRNA-like target sites in 3′ UTRs of mRNAs using their seed sequence. These interactions with transcripts other than the intended target are called “off-target”, and can involve undesirable transcript degradation and transcriptional/translational repression [13]. miRNA-like off-target effects are highly problematic in large-scale RNAi screening approaches, and many false positive hits are caused by off-target effects. Notably, since siRNAs are used widely for therapeutics as well as crop protection purposes, their miRNA-like off-target effects need to be minimized [14]. Thus, it is of major importance to understand the characteristics of the functional targets for siRNAs and enable their efficient prediction.

In plants, sRNAs are involved in reproductive transitions, for example, meiosis and gametogenesis, and regulate important epigenetic mechanisms, including genomic imprinting and paramutation. The main small-RNA classes are miRNAs, 21–22 nt long secondary siRNAs, and 24 nt long heterochromatic siRNAs (hetsiRNAs). All sRNAs in plants are modified at their 3′-end by 2′-O-methylation. This modification, nonexistent in animals, offers stability and protects sRNAs from degradation [15]. In *Arabidopsis thaliana*, 10 different AGO proteins are known to mediate the effects of several distinct types of sRNAs [16]. The movement of sRNAs in plants can be either short-range (cell-to-cell), or long-range (systemic) [17]. RNA silencing also spreads systemically over long distances in the course of days [18,19].

There is evidence that supports the existence of highly complex pathways for miRNA biogenesis and miRNA-mediated gene regulation in both animals and plants [20]. The commonalities and differences between animal and plant miRNA have been described in previous reviews in detail [20,21,22]. These differences can implicate the task of computational target prediction and thus should be accounted for. The summary of main commonalities and differences is shown in Table 1.

In bacteria, trans-acting sRNAs are a major and heterogeneous class of regulators of post-transcriptional gene expression and are often associated with chaperon proteins such as Hfq or ProQ [23,24]. They are 50–530 nt long and they regulate their mRNA targets in a (usually incomplete) base pairing-dependent manner that entails altering the mRNA translation or stability [25,26,27,28]. Even though base pairing mostly involves the 5′- and 3′-UTR regions of the target mRNA, it can also involve sites of the mRNA coding region. Bacterial sRNAs interact with their targets near the ribosomal binding site (RBS), thus repressing translation by masking the RBS or by inducing translation by making the RBS accessible [29]. Antisense sRNAs are another class of bacterial sRNAs that are cis-encoded on the opposite strand of their target gene, thus are fully complementary to their mRNA target [30]. Unlike miRNA, sRNAs may cause both up- and downregulation of their targets [31].

tRNA-derived fragments (tRFs, alternatively transfer-RNA-derived small RNAs, tsRNAs) are an emerging class of evolutionary conserved functional non-coding RNAs found across all kingdoms of life [32,33,34,35]. They are 14–32 nt long and their biogenesis involves cleavage of tRNAs precursors or mature tRNAs at specific loci and subsequent processing [36,37]. They have been previously classified into five categories: (1) tRF-5s, derived from the 5′ ends of mature tRNAs; (2) tRF-3s, from the 3′ ends of mature tRNAs with 3′-CCA termini; (3) i-tRFs, derived from cleavage of mature tRNAs; (4) tRF-1s, from the 3′ flanking sequences of pre-tRNAs with PolyU residues; and (5) tiRNAs, which are halves of tRNA cleaved at the anticodon [32,33,34,38]. The binding of tRFs to Argonaute proteins and Argonaute-mediated post-transcriptional silencing in an RNAi-like fashion have been reported only in eukaryotes [39]. Analysis of deep sequencing and AGO PAR-CLIP of sRNAs has shown that numerous reads can be mapped to tRFs, and that tRF-5s and tRF-3s can interact with target mRNAs in a fashion similar to miRNAs with 6-mers complementary to the seed [36,39,40,41]. It is found that in human HEK293 cells tRFs associate with AGO 1, 3, and 4, but not AGO 2, which is the main effector protein of miRNA function [39,42]. tRFs have also been indicated as cancer biomarkers [43,44].

Ribosomal RNAs (rRNAs) are the most abundant cellular RNA species that are the source of non-randomly generated fragments, namely, rRFs (ribosomal RNA Fragments). rRFs are an emerging class of regulators of gene expression. In plants, it has been shown that 5.8S rRF is involved in the cleavage of RPS13 and RPL5P mRNAs [45]. In H1299 cells, knocking down a 20-nt rRF-induced apoptosis, inhibited cell proliferation and led to a decrease in G2 phase cells [46]. In HeLa cells, overexpressing an rRF from the 5′ end of 28S rRNA led to the inhibition of several ribosomal proteins [47]. In Drosophila, it was shown that rRFs exhibit age-dependent Argonaute loading, comparable to that of miRNAs and tRFs [48]. Last year, a computational meta-analysis of ribosomal RNA fragments [49] from Ago1 CLASH in human showed that guanine-rich rRFs were preferentially cut in single-stranded regions of mature rRNAs between pyrimidines and adenosine, and non-randomly paired with cellular transcripts in crosslinked chimeras. In addition, numerous identical rRFs were found in the cytoplasm and nucleus in mouse Ago2-IP.

Figure 1 illustrates the mechanisms of the RNAi pathway for the various types of sRNAs. Our review focuses on the paths that involve the RISC complex; that is, the AGO-mediated silencing of the target.

Machine learning (ML) is the field of study that enables computers to learn without being explicitly programmed [50]. ML-based methods use data in order to build models, discover statistically significant patterns and relationships, and consequently make predictions on novel data [51]. One of the drawbacks of classical ML methods is their inability to work with raw data. Instead, they require a domain expert to design a feature extractor that transforms the raw data into a suitable internal representation or feature vector from which the ML method can detect or classify patterns in the input [52]. Deep Learning (DL) is a subfield of ML that essentially encompasses a class of large artificial neural networks. DL methods are able to process raw input data by constructing simple but non-linear modules; each of them transforms the representation at one level (starting with the raw input) into a representation at a higher, slightly more abstract level. With the composition of enough such transformations, very complex functions can be learned. DL-based methods have been shown to be effective for classification tasks in domains with complex feature representation [52].

To assess any classification task, there is a need for useful metrics. The most common ones are sensitivity (also called recall), accuracy, precision, and F1 score.
(1)$$Sensitivity\;\left(Recall\right)=\frac{TP}{TP+FN},$$
(2)$$Accuracy=\frac{TN+TP}{TP+FP+TN+FN},$$
(3)$$Precision=\frac{TP}{TP+FP},$$
(4)$$F1\;score=2\frac{Precision*Recall}{Precision+Recall},$$
where TP and FP are the numbers of true positive and false positive assessments, and TN and FN are the numbers of true negative and false negative assessments, respectively. Another useful metric is the Precision–Recall Area Under the Curve, PR AUC, which is the area under the Precision–Recall curve. The Precision–Recall curve is constructed by calculating and plotting the precision against the recall at a variety of thresholds. The higher the PR AUC, the better the performance of the classifier at distinguishing the positive class from the negative class.

In this paper, we present an in-depth review on the current state of the sRNA target prediction. We discuss the basic principles of experimental methods and then we focus on computational tools. For easier navigation through the review, the various tools are grouped based on the type of sRNA they are designed for. In the case of miRNAs, we focus only on the DL methods, since they are the current state-of-the-art in the field and target interactions the majority of the previous tools have already been described in numerous reviews.

## 2. Materials and Methods

### 2.1. Experimental Identification of sRNA–Target Interactions

Elucidating the interactions of sRNAs with their targets is pivotal for diagnostics and therapeutics. Even though bioinformatic approaches are the most widely used for the exploration of miRNA targets, they produce a non-negligible number of false positives [53]. Furthermore, miRNAs might interact through “non-canonical” binding sites [54,55], or with non-coding RNAs [56]. Therefore, it is essential that the small RNA–target interactions are experimentally validated.

There are three main categories of methods for the isolation and the identification of miRNA targets: (1) gene expression profiling methods, (2) immunoprecipitation methods, and (3) pull-down methods. An extensive review of the characteristics of each strategy is presented in [57] and in [58]. Methods of category (1) rely on the core of the miRNA regulatory function; that is, the mediation of mRNA degradation or repression of mRNA translation. Overexpression or inhibition of specific miRNAs and screening the subsequent response in the expression levels of genes can indicate the mRNA targets. Luciferase reporter screening can identify direct targets for miRNAs, but is limited by the availability of 3′-UTR libraries and is low throughput. The gene of interest is fused at the 3′-UTR with a luciferase reporter gene and cotransfected with a query miRNA. Targeting is measured as the differential light emission between the target gene fused with the luciferase reporter gene and a non-targeted luciferase reporter [59,60,61]. Although a set of quantitative, high-throughput methods have been developed [62,63,64], they cannot distinguish between primary and secondary miRNA targets, and they suffer from high false positive and false negative rates.

Category (2) of methods is based on immunoprecipitation (IP) of RISC proteins via specific antibody, isolation, and identification of the bound mRNA [65]. miRNA targets can thus be indirectly mapped utilizing bioinformatic tools. Crosslinking and immunoprecipitation (CLIP) methods improve the capture efficiency of IP by utilizing UV irradiation to produce covalently bound AGO–miRNA and AGO–target pairs [66]. HITS-CLIP and iCLIP can identify cross-link sites with nucleotide resolution [40,67], while the development of enhanced CLIP (eCLIP) [68] significantly improved the rate of success at generating libraries with high usable read fractions. Chimeric eCLIP, a method presented earlier this year, implements a chimeric ligation step into a simplified AGO2 eCLIP and reports up to 175-fold increased yield of recovered miRNA:mRNA interactions [69]. IP and CLIP methods provide only indirect evidence for the miRNA–target interaction. This limitation is addressed by the Crosslinking, Ligation, and Sequencing of Hybrids (CLASH) method [54,70], which ligates the miRNA and its target. Overall, immunoprecipitation methods are inherently limited by the specificity of the antibody and are of low efficiency.

Lastly, pull-down methods utilize tagged miRNA as probes to directly isolate miRNA-associated targets. These methods include the use of 3′-biotinylated RNA probes to capture miRNA targets [71,72]. Since this type of probe hinders the incorporation of miRNA into RISC, the miR-CLIP method was proposed. miR-CLIP combines miR-106a mimic probe with biotin modification and photo-reactive molecule modification at middle sites [56]. The probe cross-links to target miRNA and is subsequently immunoprecipitated with AGO2 antibody. This method is not universal, has low efficiency, and is limited by the specificity of the antibody. Photoclickable miRNA provides a universal strategy for tagging a variety of miRNAs and preserves the miRNA function within the cells [73]. The method is based on the attachment of a biotin handle through tetrazole-alkene photoclick reaction [74] to complexes containing photoclickable miRNA.

### 2.2. Computational Identification of sRNA–Target Interactions

The arsenal of methods for the prediction of sRNA targets is being enriched at a fast pace, following the advancements in the experimental techniques and in computational power. Here, we present an overview of the evolution of the computational methods in tandem with important experimental landmarks. We summarize the wide range of underlying computational techniques, and we subsequently present the computational tools developed in the last decade. The tools are arranged according to the type of sRNA molecule for which the target is to be predicted. A brief explanation of the function and/or the aim is provided for each individual tool. The list of the tools, dating from 2010 till now, along with the date of publication and their repository/web interface/source code (if available) is presented in Table 2.

#### 2.2.1. Evolution of the Methods for Computational Identification of sRNA–Target Interaction

The development of computational methods for the prediction of sRNA targets followed closely the advancements in experimental techniques. The first methods for computational prediction of miRNA targets appeared in 2003, shortly after it was suggested that miRNAs are widespread and abundant in cells [149,150,151]. In 2009, a review article [152] highlighted the tools used for human and mouse miRNA target prediction. The sequence alignment of the miRNA seed to the 3′-UTR of candidate target genes was used as the main prediction feature in the majority of the reported methods. The most commonly used tools till then were DIANA-microT 3.0 [153], ElMMo [154], miRanda [155], miRBase [156,157], PicTar [158], PITA [159], RNA22 [160], and TargetScan 5.0 [124,161]. They use heuristic algorithms based on rule matching, following the discoveries in experimental identification of miRNA targets.

However, the complexity of sRNA:mRNA interactions is the stumbling block for the heuristic methods. The last decade was a turning point for the development of target prediction tools based on machine learning (ML) and deep learning (DL) [58,162,163,164,165,166] that gradually move away from describing each condition necessary to predict a functional target toward leveraging the power of data. Some tools are still focused on a smaller number of carefully curated features based on biological findings (mirMap [139], STarMir [126]), but others generate a large number of features and let the ML method pick the best ones and discover the right way to combine them to obtain the correct prediction (TargetSpy [148], miREE [145], MultiMiTar [146], RFMirTarget [134], mirMark [127], MBSTAR [125], and miRTPred [100]). sRNA target prediction can be based on thermodynamic calculations of the sRNA-putative target hybrid (sTarPicker [89], RNApredator [88], IntaRNAv2.0 [84], and TargetRNA2 [85]), or account for multiple additional features, such as seed interaction (IntaRNAv2.0 [84]), or secondary structure (TargetRNA2 [85]). Other methods implement phylogenetic conservation (CopraRNA [86,87]). The scope of the prediction can extend from specific target site to genome-wide (sTarPicker [89], IntaRNAv2.0 [84], TargetRNA2 [85], sRNARFTarget [81], MIRZA-G [79], and RIsearch2 [78]).

Computational methods for the identification of sRNA–target interaction use a large variety of ML algorithms, and there seems to be no clear consensus as to which is the most suitable for this task. miREE uses a genetic algorithm to generate a set of sequences, which are then fed to a Support Vector Machine (SVM) algorithm. MultiMiTar also uses SVM but in combination with Multi-Objective Simulated Annealing (AMOSA) [167] to select biologically relevant features. chimiRic uses two SVM models, one for local and one for global context. mirMark evaluated several different algorithms on more than 700 features and Gaussian SVM and Random Forest (RF) performed the best. RF algorithm is also used by RFMirTarget, MBSTAR, and sRNARFTarget [81]. The miRTPred method uses the weighted voting ensemble approach, combining the predictions of the best-performing traditional and classical ensemble ML algorithms.

The mentioned supervised learning methods are based on labeled training samples and their success relies on extracting the effective sequence features that are capable of differentiating the positive and negative sRNA-gene association samples. The shortage of reliable negative sRNA-gene samples can be limiting their power. Unlike supervised learning methods, recommendation algorithms do not require negative samples. miRTRS [99] and miRTMC [101] predict miRNA targets based on a collaborative filtering recommendation algorithm. The biologically experimentally validated miRNA targets are used to construct a heterogeneous network and the miRNA–gene interaction that is not experimentally validated is predicted by filling out the unknown elements in the miRNA–gene interaction matrix.

There seems to be a poor agreement between the results of different algorithms, yet they achieve similar performance. Several tools are based on the integration of predictions from different algorithms (RFMirTarget [134], RPmirDIP [97], BCmicrO [138], and SPOT [82]). Authors claim that different algorithms rely on different mechanisms in making predictions, each of which has its own advantages, and it can be desirable to integrate their results. The RFMirTarget method improves the predictions produced by miRanda [155] using an additional 34 sequence-based features in an RF model. The RPmirDIP method uses the Reciprocal Perspective (RP) method [168] to refine predictions stored in the mirDIP database [169]. BCmicrO uses Bayesian Network to refine the prediction scores produced by TargetScan, miRanda, PicTar, mirTarget, PITA, and Diana-microT. SPOT is a pipeline that incorporates sTarPicker [89], TargetRNA2 [85], IntaRNA [170], and CopraRNA [86,87].

Following the ML algorithms, a variety of neural network architectures has been utilized for the task of sRNA target prediction. The utilization of artificial neural networks (ANNs) can be traced back to the year 2010 when MTar [142] used a feed-forward three-layer multi-layer perceptron (MLP) for the classification of target sites. Other early adopters of artificial neural networks were DIANA-microT-ANN [140] and HomoTarget [135]. DIANA-microT-ANN used a recurrent neural network (RNN) with two layers to combine predictions from all candidate target sites (CTSs) in the mRNA to obtain a final prediction. The year after, HomoTarget introduced the Pattern Recognition Neural Network (PRNN) for predicting miRNA targets based on manually extracted features.

The shift towards deep learning (DL) methods started around the year 2016. The main reason for this was the critique of the huge bias introduced by the manual feature crafting and selection. Even though DL methods are capable of extracting important features directly from the raw input, the first DL methods were still using handcrafted features (MiRTDL [117], DeepMirTar [106]) and neural networks were used only to make better predictions from these features. A later method, miTarDigger [102], used hand-crafted structural features together with raw sequence. Subsequent methods moved away from manual feature crafting and tried to work directly with raw data in the sequence format (deepTarget [119], miRAW [107], cnnMirTarget [98], miTAR [94], TargetNet [92], and DMISO [90]). However, miRAW used additional features (binding stability and site accessibility) in the a posteriori filtering step to improve predictions.

As mentioned before, the first common architecture in the field of sRNA target prediction is MLP, used by MTar and miRAW. Another common architecture is the convolutional neural network (CNN) [171] that is used by MiRTDL, cnnMirTarget, and other methods that combine convolutional layers with other architectures. TargetNet uses the ResNet [172] architecture that introduces residual connections on top of convolutional layers. MiTAR and DMISO use CNN followed by recurrent neural network (RNN), precisely, Long Short–Term Memory (LSTM) neural network [173] arguing that this hybrid architecture is better suited for extracting sequential and spatial features from sRNA and mRNA [98,174].

The last commonly used architecture is the autoencoder [175]. DeepTarget uses autoencoder together with RNN, DeepMirTar uses stacked denoising autoencoders, and miTarDigger adds convolutional denoising autoencoders to the stacked denoising autoencoders. The advantage of autoencoders is that they can be pre-trained in an unsupervised manner—the objective is to learn a meaningful encoding of the input sequence and then reconstruct the sequence from the encoding.

#### 2.2.2. Description of Selected Computational Methods

Computational prediction of sRNA–target interactions is a highly active field of research that has produced tens of prediction methods during the last decade. Most tools focus on miRNA target prediction, a fraction of them predicts targets of a variety of sRNAs, and a couple of tools enable the prediction of tRF targets (tRFTars [76], tRFTar [75]). Moreover, a few methods have been developed for the prediction of siRNA off-targets, namely, MIRZA-G [79], RIsearch2 [78], and siRNA-Finder [77]. Given the fact that the vast majority of miRNA target prediction tools have already been described in several reviews published in the last two years [58,165,176], and that DL methods are the current state-of-the-art in the field, from the numerous miRNA target prediction tools we describe only those that are based on DL methods. An overview of the tools for the prediction of the target of the various types of sRNAs is presented in Table 2.

##### sRNA–Target Interactions

sTarPicker [89] is an ensemble classifier trained on 32 experimentally verified bacterial sRNA–mRNA repression pairs from sRNATarBase 1.0 [177,178]. Hybridization between an sRNA and an mRNA target is based on a two-step model: (1) seed matching between the sRNA and a target, and (2) elongation of the hybrid so that the duplex formed is stable. The hybridization is assessed by ${\Delta G}_{\mathrm{open}}$ (free energy), ${\Delta G}_{\mathrm{hybrid}}$ (computed by RNAduplex of Vienna RNA package [179]), and ΔΔG that indicate thermodynamic stability and site accessibility. sTarPicker picks stable seeds based on rules constructed from known seed bindings of 17 pairs, extends the binding sites by 100 nt upstream and downstream of the seed, extracts the features of the binding sites, and the ensemble classifier predicts the probability of the sRNA–target interaction. sTarPicker is no longer available online.

In the same year, RNApredator [88] became available as a web server for the prediction of bacterial sRNA targets. After an input sequence is submitted, its targets can be searched against a set of over 2155 genomes and plasmids from 11,183 bacterial species. The output contains a table of the 100 most stable duplexes predicted by the dynamic programming approach RNAplex [180], hybridization energy, and structure in dot-bracket notation. Additional features such as enrichment in Gene Ontology terms, target site accessibility, and cellular pathways can be obtained in an automatic post-processing step.

CopraRNA (Comparative Prediction Algorithm for sRNA Targets) [86,87] integrates phylogenetic information to predict sRNA targets on the genomic scale for a set of given organisms. It employs a statistical model and computes whole genome target predictions based on whole genome target screens for homologous sRNAs performed by IntaRNA [170]. The method aims to address the high false positive rate (FFP) of previous approaches relying on thermodynamic models (including RNApredator), base complementarity, or seed conservation. It combines individual *p*-values among clusters of genes predicted by IntaRNA to generate a weighted *p*-value and false discovery rate (FDR)-corrected q-value. CopraRNA reconstructs regulatory networks upon functional enrichment (using the DAVID database [181]) and network analysis, and predicts the sRNA domains for target recognition and interaction. CopraRNA is available at the Freiburg RNA tools webserver [182], requires an input in FASTA format with its RefSeq ID, and allows for the visualization of interacting regions. The method requires the conservation of both sRNA and mRNA in a minimum of four bacterial species, which renders it unsuitable for species-specific sRNA target prediction.

TargetRNA2 [85] is a web server that identifies mRNA targets upon being given an sRNA sequence and the name of a bacterial replicon. The prediction calculates a variety of features by means of previously published methods: conservation (calculated utilizing BLASTN [183] and ClustalW2 [184]) and secondary structure (using RNAfold from Vienna RNA Package) of the sRNA. Additional features are the accessibility of regions in the mRNA secondary structure (calculated based on RNAplfold [185]) and the sRNA-putative target hybridization energy (based on calculations by RNAduplex from Vienna RNA Package). TargetRNA2 produces *p*-values for predicted interactions based on the hybridization energy scores of a randomized mRNA pool. If available, the method can integrate RNA-seq data and consider co-differential gene expression. TargetRNA2 scans for sRNA–mRNA interactions around the 5′-UTR of the mRNA or proximate to the beginning of the mRNA coding sequence.

IntaRNAv2.0 [84] is the open-source reimplementation of IntaRNA [170] that favors seed interactions. It allows for user selection of energy parameters, seed constraints, and accessibility computation. The user can submit either a list of putative interacting RNA pairs to perform an all-versus-all prediction, or a single RNA so as to perform a genome-wide target screen. The method produces *p*-values based on the transformation of the energy scores calculated for all putative target binding sites with non-positive energy scores. The web server offers visualization of minimal energy profiles of interacting RNAs, thus enabling the study of alternative RNA–RNA interactions and the analysis of mutational effects.

psRNATarget [83] is a web server for the identification of target genes of plant miRNAs. The user can submit either (i) a list of sRNAs to search against preloaded target transcript libraries, (ii) candidate target transcripts to search against sRNAs from miRBase, or (iii) candidate sRNA–mRNA pairs. The procedure consists of two steps: first analysis of the sRNA–target mRNA complementary matching based on a scoring schema; and second, evaluation of the target site accessibility. It allows for customization of the scoring and search for both canonical and non-canonical targets. Mismatches in the mismatch-sensitive seed region are penalized more than the positive contribution of the complementary base pairing. The seed region (in the original version [186], defined as being in vertebrates, nucleotides 2–7 [11]) has been extended to nucleotides 2–13, allowing for up to 2 mismatches, according to the plant miRNA target recognition patterns [187]. psRNATarget is no longer available online.

In order to provide a collated and standardized result report, SPOT (sRNA Target Prediction Organizing Tool) [82] implements multiple algorithms for sRNA target prediction. SPOT is a pipeline that incorporates sTarPicker [89], TargetRNA2 [85], IntaRNA [170], and CopraRNA [86,87]. The minimal input consists of an sRNA sequence in FASTA format and the RefSeq ID of the target genome. To include CopraRNA, homologous sRNAs, and additional RefSeq IDs should be included. The interface allows for parameter setting for each method and for filtering the results. The utility and sensitivity of the pipeline were tested on two well-characterized *E. coli* sRNA models, SgrS [188] and RyhB [189]. Using more stringent parameters (stricter significance thresholds and smaller search windows upstream/downstream of start codons), or combining more than three algorithms for the prediction, decreases the FFP at the cost of sensitivity. When at least two methods converge on a prediction for those datasets, SPOT achieves sensitivity ≥ 75% and FFP ≤ 50%.

sRNARFTarget [81] is a machine-learning-based method for transcriptome-wide sRNA target prediction. It utilizes a random forest (RF) trained on the trinucleotide frequency difference between 745 sRNA–mRNA pairs from 37 bacterial species obtained by RNA-seq [190], MAPS [191], GRIL-seq [192], RIL-seq [193], and CLASH [194]. Added information on the predicted secondary structure was not proven to improve the overall performance. It outperforms IntaRNAv2.0 in accuracy, running time, and ranking of true interacting pairs. However, CopraRNA is a more suitable option for prediction when sRNA homologs are available, as it was shown to outperform both methods in accuracy. The versatility and usability of the method on any sRNA–target pair rely on the fact that the prediction depends solely on the sequence.

##### miRNA–Target Interactions

cnnMirTarget [98] employs a CNN to automatically integrate the patterns in the raw sequence data, avoiding the hand-crafted selection of features. The tool predicts the target gene of miRNAs through scanning the full length of gene transcripts. cnnMirTarget is trained on a positive dataset constructed from the three sources: CLASH [54], AGO-CLIP [195], and MirTarBase [196,197], and negative data generated by pseudo combinations of miRNA and gene omitting the miRNA:mRNA pairs in MirTarBase. The trained model is evaluated on both site-level and gene-level data, the latter of which was downloaded from MirTarBase and Diana TarBase [198].

miTarDigger [102] utilizes two types of neural architectures: stacked denoising autoencoders (SDA) [199] and convolutional denoising autoencoders (CAE) [200]. Each type has its own functions: SDA is used to process sequence and CAE to process structure features. The results of two encoders are fused and then fed into the fully connected network and a logistic regression layer. Most of the existing studies have not considered the impact of upstream and downstream sequences of target sites on the prediction results and miTarDigger is exploiting this gap. miTarDigger is trained on CLASH data [54]; hence, it predicts on the site level. To be able to perform gene-level predictions, miTarDigger finds all CTSs in a given mRNA utilizing miRanda software [201].

RNNs and MLPs might be unsuitable for the task or miRNA target prediction, as they may not be able to efficiently capture the spatial and sequential features of the miRNA:target hybrid. miTAR [94] uses a combination of CNN and RNN architecture, exploiting the fact that CNNs excel in learning spatial features and RNNs discern sequential features [163,174]. The miTAR neural network is trained on site-level data obtained from miRAW [107] and DeepMirTar [106].

Predicting a functional miRNA:CTS pair from sequence only is not enough to fully capitalize the information underlying miRNA–CTS interactions. TargetNet [92] addresses this issue by proposing a novel miRNA:CTS encoding. Previous methods use one-hot encoding to convert only sequences into numerical representations. In contrast, TargetNet incorporates additional information on how the extended seed regions of a miRNA:CTS pair are aligned and form binding. The result of this encoding is a 2D matrix that is processed by a deep residual network (ResNet) with 1D convolutions.

##### tRF–Target Interactions

The predictive power of methods aimed at miRNA targets is poor for tRF targets [202,203]. tRFTars [76] offers the first database for predicting potential targets of tRFs in human, and is available as a web interface. It utilizes a Genetic Algorithm (GA) to select features of tRF–mRNA pairs, and Support Vector Machine (SVM) to build prediction models for tRF targets. The method utilized interacting pairs identified in AGO complexes by CLASH [54] in HEK293 cells and CLEAR-CLIP [204] in Huh-7.5 cells, and mRNA sequences from UCSC 2019 [205]. After preprocessing and filtering the data, the method was trained on 547 positive pairs (489 tRF-3 and 58 tRF-5 pairs) and 2000 negative pairs (1596 tRF-3 and 404 tRF-5 pairs). Feature assessment showed significantly different features of sequences involved in mRNA targeting and in the background. The most significant features found are minimum free folding energy (MFE), position 8 match, number of bases paired in the tRF–mRNA duplex, and length of the tRF, in agreement with previous studies [39,40]. The trained GA-SVM models were shown to outperform the intersection of the miRNA target prediction models TargetScan [124,161] and miRanda [155].

tRFTar [75] is a publicly accessible multi-functional platform that contains 920,690 interactions between 12,102 tRFs and 5688 target genes identified by CLIP-seq in human. The authors utilized data from human 160 Ago CLIP (HITS-CLIP and PAR-CLIP), as well as annotation of 26,744 tRFs from MINTBase v2.0 [206] and genomic annotation from RefSeq [207]. After preprocessing, tRF–target interactions were predicted based on the MFE of the duplexes and simulated annealing was performed by RNAduplex [179]. Only those duplexes that met the normalized MFE threshold were retained and further validated by datasets from 6 CLASH experiments. A tRF-gene co-expression profile was constructed indicating context-dependent regulatory functions. 5′-tRFs and 3′-tRFs were found to be more likely candidates for AGO-mediated gene expression regulation, and their interaction sites tend to be preferentially distributed. The tRFTar platform allows for the custom search of interactions, genome browser, GO enrichment, and co-expressed interaction filtering.

##### siRNA Off-Target Interactions

MIRZA-G [79] is a suite of algorithms (currently the web server is not accessible) for the genome-wide prediction of (non-) canonical miRNA targets and siRNA off-targets. It implements the MIRZA [132] biophysical model for the prediction of the miRNA–target interaction energy. The model, trained and evaluated on data from a set of 26 experiments on humans, considers features such as nucleotide composition around putative target sites, their structural accessibility, and location within 3′ UTRs. Adding evolutionary conservation as a feature improves the prediction of siRNA target sites, in accordance with previous findings [208].

RIsearch2 [78] uses a single integrated seed-and-extend framework based on suffix arrays to predict RNA–RNA interactions. Unlike its first version [209], RIsearch2 follows the two-stage strategy of the seed-and-extend paradigm. In the first step, it uses suffix arrays to locate maximal stretches of perfect complementarity (wobble pairs allowed), and in the second step extends those seed matches on either end using dynamic programming with the scoring scheme introduced in [209]. The study also presents a pipeline that predicts siRNA off-target transcripts, and the off-targeting potential for a given siRNA based on genome-wide RIsearch2 predictions combined with target site accessibilities and transcript abundance estimates. The pipeline accounts for intramolecular interactions of the targeted transcripts, and allows for user-defined seed and extension constraints. Rlsearch2, originally constructed for predictions on humans, can be tuned for any other organism.

siRNA-Finder [77] is a tool for the prediction of RNAi sequences and off-target search in plants, designed for MS Windows. It utilizes the BOWTIE-based sequence similarity search for putative siRNA targets, the probability calculation of local target-site accessibility, and thermodynamics—as well as a sequence-based prediction for strand selection. It includes two pipelines with different functionalities: High Sensitivity for off-target search, and High Efficiency for RNAi-construct design.

cWords [80] is a tool based on rigorous statistical methods, designed to extract correlations of differential expression and motif occurrences. The method can assist the exploratory analysis of enriched words and degenerate motifs such as noncanonical miRNA-binding sites and RNA-binding protein binding sites by providing methods for clustering and visualization of enriched words with similar sequences. It has been demonstrated that cWords, originally designed for miRNAs, can also be used for the identification of potential siRNA off-target binding.

## 3. Discussion

The advancements in high-throughput experimental techniques and the increasing computational power have propelled forward the development of computational methods for the prediction of sRNA targets. A plethora of methods with a variety of features and assumptions is available for diverse types of sRNAs. The emerging field of Deep Learning has offered current-state-of-the-art methods for sRNA target prediction; nevertheless, there are still multiple challenges to be addressed, and limitations of existing methods that still leave the field open for further development.

An important distinction to be made for the various sRNA target prediction methods presented here is the goal of the prediction. Older methods (e.g., TargetScan, microT, and others) attempted to predict targeting at the miRNA:gene level. Usually, such methods would use one model for the identification of putative target sites, and a different model that would combine such target sites into a gene-level prediction. This functionality could then be evaluated on experimental datasets coming from overexpression or knockout of specific miRNAs, and prediction of the effects on their targets. Later Deep Learning-based methods have by and large been trained on target-site level data (validated luciferase targets, CLASH, and similar). The majority of the methods consider a miRNA:gene pair to be functional when at least one functional site is present in a target gene (cnnMirTarget, miTAR, TargetNet, and DMISO), whereas some others consider additional features in the post-processing, such as binding stability and site accessibility (miRAW). We notice that many methods that are currently trained and evaluated on target-site data, use direct comparisons with older methods (commonly TargetScan) that are trained on a completely different task, namely, sRNA: gene-level prediction. These comparisons are summarized in Table 3. We would like to caution against such comparisons as non-informative for the user and the field at large. We consider it to be beneficial for subsequent target prediction programs to clarify the distinction and the goal of their program, and to benchmark against programs of the same category, on benchmarks that are relevant to the task.

Converting site-level predictions to gene-level predictions is not a trivial task. In brief, when scanning a transcriptome for candidate targets of a specific sRNA, one expects to encounter orders of magnitude more non-target sites, than bona fide target sites. Such a substantial class imbalance is another challenge that sRNA target prediction methods, and especially DL methods, have to deal with, since they perform best when classes are balanced. All of the DL methods we presented here are trained on target site-level data, often balanced in number between positive and negative samples, but the ultimate goal is to predict interactions on the gene level, which exhibits an order of magnitude class imbalance. Typically, the whole mRNA is split into smaller parts representing CTSs; however, only a fraction of them contains real target sites. A common approach involves filtering out CTSs that have a low probability of containing real target sites. Rules for filtering are based on current experimental findings about interactions between miRNA and its target: extended loose seed matching (deepTarget, TargetNet), the potential to create a stable duplex (cnnMirTarget), or both types of these constraints (miRAW), thus introducing an undesirable bias as the filtering is performed using handcrafted conditions. MiRAW claims that CTS pre-filtering is required when there are too few samples for the neural network to learn all necessary features and handle pre-filtering itself. Recently, TargetNet investigated the effect of CTS pre-filtering on the classification performance and obtained a similar F1 score with and without CTS pre-filtering, with the only difference being the computational time. At this point, the transition from the target site to target gene-level prediction is still unclear and based on heuristics. We believe that there is a need for a new method that systematically combines target site-level predictions into gene-level predictions using contemporary machine learning methods. When working with computationally predicted targets, it is important to remember that thorough experimental validation is paramount, since all current target prediction programs will produce false positive targets, as well as numerous false negative targets. Until the exact rules of small RNA targeting are fully understood, we need to treat all predictions as educated guesses until thoroughly validated.

Obtaining larger and more reliable experimental datasets is of utmost importance for the development of efficient computational methods that can, in turn, facilitate the experimental validation of sRNA targets. Site-level CLASH data have offered the direct identification of sRNA targets, and a recent modification of chimeric eCLIP promises the recovery of miRNA:mRNA interactions by 70-fold. Still, obtaining large-scale good quality gene-level experiments remains a challenge that can resolve the artificial assumptions for the computational prediction of sRNA targets. An additional challenge is that available datasets are quickly used by computational methods for training, making true benchmarking against never-before-seen data impossible, unless someone performs dedicated overexpression or knock-out experiments for benchmarking. We would suggest the future development and publication of dedicated benchmark datasets for both the target site level (based on new CLASH or chimeric eCLIP data) and at the target gene level (based on overexpression or knockout experiments). These benchmarks must be developed explicitly to solve the fragmentation of the field, and have exact testing sets that cannot be used for training future methods, so as to allow for continuous testing against them. In the current state of the field, meaningful benchmarking of tools is impossible unless new experiments are performed.

Finally, the field of sRNA target prediction is currently dominated by miRNA target prediction, with a small spread into tRFs and other small RNA species.

The exploration of the regulatory role of other sRNAs, including rRFs, still remains to be further elucidated, and computational methods for the prediction of their targets are yet to be developed. As was previously shown, the prediction of reliable RNA–RNA interactions can be used to infer the functional relationships of miRNAs [216]. Acquiring such interaction data can accelerate the discovery of new ncRNAs and provide insight into their involvement in regulating cellular output. We believe that exploring the potential for regulatory roles of other non-coding RNA families will be an important development in the field in the next years, as long as it is supported by high-quality experimental data and benchmarks.

To conclude, the sRNA target prediction field has seen great development in the past 5 years with the advent of Deep Learning methods and new experimental datasets such as CLASH, but no newly produced method has decisively outperformed others either in the target-site prediction level, or the target-gene prediction level. The field remains open to further developments, as the need for new and properly set up experimental benchmarks increases.

## 4. Conclusions

RNA interference is a widespread, evolutionary conserved mechanism that is of great significance for the fields of therapeutics and diagnostics. At its core, it is driven by small RNAs that target mRNAs for cleavage or translational repression. The recent advancements in high-throughput sequencing techniques, in tandem with the rapidly developing field of Machine Learning, have shed light on these complex RNA–RNA interactions, and have produced numerous computational methods for the prediction of sRNAs targets. However, the field of sRNA target prediction is currently dominated by miRNA target prediction, with a small spread into tRFs and other small RNA species. In this review, we document the development of ML and other computational methods for the prediction of small RNA targets, with emphasis on the non-miRNA sRNAs, and we highlight the limitations and the future prospects of the research in the field. Additionally, we provide a brief overview of the high-throughput methods utilized for the detection of RNA–RNA interactions.

## Figures and Tables

**Figure 1 biology-11-01798-f001:**
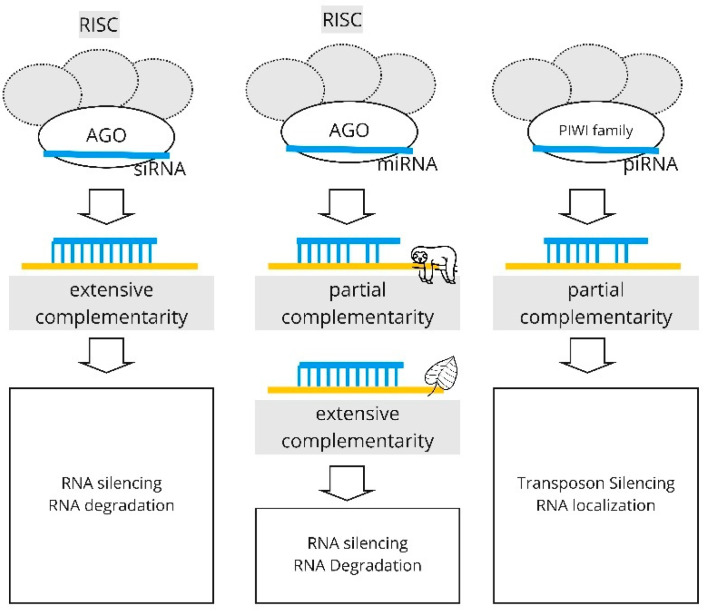
The pathways of RNA interference for the various types of sRNAs. The miRNA pathway in animals involves partial complementarity with the target; in plants, the complementarity is extensive.

**Table 1 biology-11-01798-t001:** Summary of commonalities and differences between animal and plant miRNA-mediated gene regulation.

Feature	Plants	Animals
Size (number of nucleotides)	18–25 nt	18–25 nt
Mechanism of target recognition	Ribonucleotide complementarity	Ribonucleotide complementarity
Location of miRNA binding sites within target mRNAs	Predominantly in the open reading frame	Predominantly 3’ untranslated region (3’UTR)
Number of miRNA binding sites within target mRNAs	Generally single	Generally multiple
miRNA–mRNA complementarity	Generally a perfect complementarity	Imperfect; seed sequences and variable flanking complementarity

**Table 2 biology-11-01798-t002:** Bioinformatical and Machine Learning methods for the prediction of sRNA targets. The last column indicates the availability of each method: o (open source), s (standalone), w (web service), - (not available/not functional).

	Method	Year	Repository/Web App	
tRFs target prediction				
1	tRFTar [75]	2021	http://www.rnanut.net/tRFTar/ (accessed on 7 December 2022)	w
2	tRFTars [76]	2021	http://trftars.cmuzhenninglab.org:3838/tar/ (accessed on 7 December 2022)	o
siRNAs off-target prediction				
3	si-Fi [77]	2019	https://github.com/snowformatics/siFi21- (accessed on 7 December 2022)	o
4	RIsearch2 [78]	2017	https://rth.dk/resources/risearch/ (accessed on 7 December 2022)	s
5	MIRZA-G [79]	2015	http://www.clipz.unibas.ch/index.php?r=tools/sub/mirza_g (accessed on 7 December 2022)	-
6	CWords [80]	2013	https://servers.binf.ku.dk/cwords/ (accessed on 7 December 2022), https://github.com/simras/cWords (accessed on 7 December 2022)	o, w
sRNAs target prediction				
7	sRNARFTarget [81]	2021	https://github.com/BioinformaticsLabAtMUN/sRNARFTarget (accessed on 7 December 2022)	o
8	SPOT [82]	2019	https://github.com/phdegnan/SPOT (accessed on 7 December 2022)	o
9	psRNATarget [83]	2018	https://www.zhaolab.org/psRNATarget/ (accessed on 7 December 2022)	w
10	IntaRNA 2.0 [84]	2017	http://www.bioinf.uni-freiburg.de/Software/ (accessed on 7 December 2022), http://rna.informatik.uni-freiburg.de/ (accessed on 7 December 2022)	w
11	TargetRNA2 [85]	2014	http://cs.wellesley.edu/~btjaden/TargetRNA2/ (accessed on 7 December 2022)	w
12	CopraRNA [86,87]	2013	http://rna.informatik.uni-freiburg.de/CopraRNA/ (accessed on 7 December 2022)	w
13	RNApredator [88]	2011	http://rna.tbi.univie.ac.at/cgi-bin/RNApredator/target_search.cgi (accessed on 7 December 2022)	w
14	sTarPicker [89]	2011	http://ccb.bmi.ac.cn/starpicker/ (accessed on 7 December 2022, )	-
miRNAs/isomiRs target prediction				
15	DMISO [90]	2022	http://hulab.ucf.edu/research/projects/DMISO/ (accessed on 7 December 2022)	s
16	SubmiRine [91]	2015	https://research.nhgri.nih.gov/software/SubmiRine/index.shtml (accessed on 7 December 2022)	o
miRNAs target prediction				
17	TargetNet [92]	2022	https://github.com/mswzeus/TargetNet (accessed on 7 December 2022)	o
18	mintRULS [93]	2022	https://zenodo.org/record/6360587#.Yy2IV9VByV4 (accessed on 7 December 2022)	o
19	miTAR [94]	2021	https://github.com/tjgu/miTAR (accessed on 7 December 2022)	o
20	SG-LSTM-FRAME [95]	2021	https://github.com/Xshelton/SG_LSTM (accessed on 7 December 2022)	o
21	miRgo [96]	2020	http://predictor.nchu.edu.tw/miRgo/index.php (accessed on 7 December 2022,)	-
22	RPmirDIP [97]	2020	https://www.cu-bic.ca/RPmirDIP (accessed on 7 December 2022,) https://dataverse.scholarsportal.info/dataset.xhtml?persistentId=doi:10.5683/SP2/LD8JKJ (accessed on 7 December 2022)	- w
23	cnnMirTarget [98]	2020	https://github.com/zhengxueming/cnnMirTarget (accessed on 7 December 2022)	o
24	miRTRS [99]	2020		-
25	miRTPred [100]	2020	http://bicresources.jcbose.ac.in/zhumur/mirtpred/ (accessed on 7 December 2022)	s
26	miRTMC [101]	2020	https://github.com/hjiangcsu/miRTMC (accessed on 7 December 2022)	o, s
27	miTarDigger [102]	2020		-
28	Min3 [103]	2019	https://sourceforge.net/projects/mirt3/ (accessed on 7 December 2022)	o
29	mirTime [104]	2019	https://github.com/mirTime/mirtime (accessed on 7 December 2022)	o
30	CCmiR [105]	2018	http://hulab.ucf.edu/research/projects/miRNA/CCmiR/ (accessed on 7 December 2022)	s
31	DeepMirTar [106]	2018	https://github.com/Bjoux2/DeepMirTar_SdA (accessed on 7 December 2022)	o
32	miRAW [107]	2018	https://bitbucket.org/account/user/bipous/projects/MIRAW (accessed on 7 December 2022)	o, s
33	MiTarget [108]	2018	http://rna-informatics.uga.edu/12_software.php (accessed on 7 December 2022)	o
34	Tiresias [109]	2018	https://bitbucket.org/cellsandmachines/tiresias-context-specific-mirna-interactome-mapping/src/master/ (accessed on 7 December 2022)	o
35	Context-MMIA [110]	2017	http://epigenomics.snu.ac.kr/contextMMIA/ (accessed on 7 December 2022)	w
36	MicroTarget [111]	2017	https://bioinformatics.cs.vt.edu/~htorkey/microTarget (accessed on 7 December 2022)	o
37	miRBShunter [112]	2017	https://github.com/TrabucchiLab/miRBShunter (accessed on 7 December 2022)	o
38	miRTar2GO [113]	2017	http://www.mirtar2go.org/ (accessed on 7 December 2022)	w
39	miRTarVis+ [114]	2017	http://hcil.snu.ac.kr/research/mirtarvisplus (accessed on 7 December 2022)	w
40	miSTAR [115]	2017	http://mi-star.org/ (accessed on 7 December 2022)	w
41	chimiRic [116]	2016	https://bitbucket.org/leslielab/chimiric/src/master/ (accessed on 7 December 2022)	o
42	MiRTDL [117]	2016	http://nclab.hit.edu.cn/ccrm (accessed on 7 December 2022 )	-
43	TargetExpress [118]	2016	http://targetexpress.ceiabreulab.org/ (accessed on 7 December 2022)	-
44	deepTarget [119]	2016	http://data.snu.ac.kr/pub/deepTarget/ (accessed on 7 December 2022)	-
45	TarPmir [120]	2016	http://hulab.ucf.edu/research/projects/miRNA/TarPmiR/ (accessed on 7 December 2022)	s
46	Avishkar [121]	2015	https://bitbucket.org/cellsandmachines/avishkar/src/master/ (accessed on 7 December 2022)	o
47	MiRNALasso [122]	2015	https://nba.uth.tmc.edu/homepage/liu/miRNALasso/ (accessed on 7 December 2022)	s
48	miRTarVis [123]	2015	http://hcil.snu.ac.kr/~rati/miRTarVis/index.html (accessed on 7 December 2022)	s
49	TargetScan v7.0 [124]	2015	https://www.targetscan.org/ (accessed on 7 December 2022)	w
50	MBSTAR [125]	2015	https://www.isical.ac.in/~bioinfo_miu/MBStar30.htm (accessed on 7 December 2022)	-
51	miRTarVis+	2017	http://hcil.snu.ac.kr/research/mirtarvisplus (accessed on 7 December 2022)	w
52	StarMir [126]	2014	https://sfold.wadsworth.org/cgi-bin/starmirtest2.pl (accessed on 7 December 2022)	w
53	mirMark [127]	2014	https://github.com/lanagarmire/MirMark (accessed on 7 December 2022)	o
54	ProMISe [128]	2014	https://bioc.ism.ac.jp/packages/3.11/bioc/html/Roleswitch.html (accessed on 7 December 2022)	o
55	TargetScore [129]	2014	http://www.bioconductor.org/packages/devel/bioc/html/TargetScore.html (accessed on 7 December 2022)	o
56	IDA approach [130]	2013	https://academic.oup.com/bioinformatics/article/29/6/765/184183#supplementary-data (accessed on 7 December 2022)	o
57	MicroMUMMIE [131]	2013	https://ohlerlab.mdc-berlin.de/files/duke/MUMMIE/download.html (accessed on 7 December 2022)	o
58	MIRZA [132]	2013	http://www.clipz.unibas.ch/downloads/mirza/ (accessed on 7 December 2022)	-
59	MREdictor [133]	2013	http://mredictor.hugef-research.org/ (accessed on 7 December 2022)	-
60	RFMirTarget [134]	2013		-
61	HomoTarget [135]	2013	http://lbb.ut.ac.ir/Download/LBBsoft/homoTarget/ (accessed on 7 December 2022)	-
62	CoSMic [136]	2012	https://www.weizmann.ac.il/complex/compphys/software/cosmic/ (accessed on 7 December 2022)	s
63	DIANA-microT-CDS [137]	2012	https://dianalab.e-ce.uth.gr/html/dianauniverse/index.php?r=microT_CDS (accessed on 7 December 2022)	w
64	BcmicrO [138]	2012	http://compgenomics.utsa.edu/gene/gene_1.php (accessed on 7 December 2022)	w
65	mirMap [139]	2012	https://mirmap.ezlab.org/ (accessed on 7 December 2022)	w
66	DIANA-microT-ANN [140]	2012	http://microrna.gr/microT-ANN (accessed on 7 December 2022)	-
67	mmPRED [141]	2012	https://bmcgenomics.biomedcentral.com/articles/10.1186/1471-2164-13-620#MOESM11 (accessed on 7 December 2022)	o
68	MTar [142]	2012		-
69	Targetprofiler [143]	2012	http://mirna.imbb.forth.gr/Targetprofiler.html (accessed on 7 December 2022)	w
70	PACMIT [144]	2011	https://paccmit.epfl.ch/ (accessed on 7 December 2022)	w
71	miREE [145]	2011		-
72	MultiMiTar [146]	2011	https://www.isical.ac.in/~bioinfo_miu/multimitar.htm (accessed on 7 December 2022)	-
73	ProbmiR [147]	2011	http://www.baskent.edu.tr/~hogul/probmir/ (accessed on 7 December 2022)	o
74	TargetSpy [148]	2010	http://webclu.bio.wzw.tum.de/targetspy/index.php (accessed on 7 December 2022)	s, w

**Table 3 biology-11-01798-t003:** Comparison of computational methods for sRNA:target prediction.

Program	Targetnet [92]	miTAR [94]	RPmirDIP [97]	miRTMC [101]	mirTarDigger [102]	cnnMirTarget [98]	miRAW [107]
PITA [159]	0.22						0.74
miRanda [155]	0.36			0.69	0.66		
mirSVR [210]							0.41
microT-CDS [211]							0.73
miRDB [212]	0.21					0.23	0.21
mirza-G [79]							0.52
Paccmit [213]							0.41
Targetscan [124]	0.47			0.67	0.62	0.31	0.56
deepTarget [119]	0.49			0.69			
TarPmiR [120]					0.78		
metaMIR [214]						0.78	
DeepMirTar [106]					0.94		
miRAW [107]	0.73	0.95					0.93
mirDIP [169]			0.88				
miRTRS [99]				0.70			
GMCLDA [215]				0.61			
cnnMirTarget [98]						0.79	
mirTarDigger [102]					0.96		
miRTMC [101]				0.72			
RPmirDIP [97]			0.93				
miTAR2 [94]		0.97					
TargetNet [92]	0.77						
	F1-score on balanced miRNA:mRNA target pairs (dataset from miRAW)	F1-score on miRAW dataset	Bootstrap testing PR AUC	AUC on different independent datasets. Showing dataset 1 (based on miRTarBase), as results are similar.	F1-score on target interactions vs. artificial miRNAs.	The experimentally validated positive dataset contains 7815 interactions; the negative dataset contains 281 pseudo-interactions.	F1-score using full testing dataset, constructed from various external sources

## Data Availability

Not applicable.

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
