# Peer review of "Small RNA Targets: Advances in Prediction Tools and High-Throughput Profiling"

_biology, 2022, doi:10.3390/biology11121798_

Round 1
Reviewer 1 Report
The authors have presented an in-depth review for the current state of the small RNA target prediction tools. I think it would improve the manuscript better if the authors can show the different mechanisms of small RNA activities via figures or diagrams. Moreover, small RNAs in different regions of life may act very differently. It is necessary that the authors provide a brief up-to-date knowledge on their activities. For examples, miRNAs in plants and animals may act very differently at post-transcriptional levels. This should be highlighted to show case the difficulties that bioinformatic tools face when they have to deal with small RNAs from different species.
Author Response
The authors have presented an in-depth review for the current state of the small RNA target prediction tools. I think it would improve the manuscript better if the authors can show the different mechanisms of small RNA activities via figures or diagrams. Moreover, small RNAs in different regions of life may act very differently. It is necessary that the authors provide a brief up-to-date knowledge on their activities. For examples, miRNAs in plants and animals may act very differently at post-transcriptional levels. This should be highlighted to show case the difficulties that bioinformatic tools face when they have to deal with small RNAs from different species.
According to the reviewer's sugestion, we added a schema (Figure 1) that illustrates the RNA interference pathway for the various types of small RNAs. We also wrote a short paragraph and compiled a table (Table 1) that highlights the important features and differences between animal and plant miRNAs.
Reviewer 2 Report
The authors have surveyed a very broad range of small RNA target prediction tools and reported in detail on the features of several of them. This review may serve as a guide to determine which tools to use when making small RNA target predictions, and may also be useful in developing successor tools for small RNA target prediction. Before publication, this review needs some minor revisions, in my opinion.
There are no sentences at the end of Section 1 (Introduction) explaining the purpose or structure of this review. These are necessary for readability.
Some explanations are redundant. For example, sRNARFtarget is explained in l.395 and l.464, which duplicates the content. It would be better to change the structure of the paper and explain it in one place.
Sec 2.3: “Databases of sRNAs and sRNA:target interactions” Unlike the description of the tools in section 2.2, I think the description of the database is too limited. Since this manuscript is a review of the sRNA target prediction tools, it would be unnecessary to include this section.
l.138: “PR AUC” It is difficult to understand this explanation of PR AUC because how the curve is drawn is not explained.
l.502: “RNA-BP binding site” What does "BP" stand for? It is probably not a very common abbreviation.
Author Response
The authors have surveyed a very broad range of small RNA target prediction tools and reported in detail on the features of several of them. This review may serve as a guide to determine which tools to use when making small RNA target predictions, and may also be useful in developing successor tools for small RNA target prediction. Before publication, this review needs some minor revisions, in my opinion.
There are no sentences at the end of Section 1 (Introduction) explaining the purpose or structure of this review. These are necessary for readability.
The following paragraph has been added to the Introduction: “In this paper, we present an in-depth review for the current state of the sRNA target prediction. We discuss the basic principles of experimental methods and then we focus on computational tools. For easier navigation through the review, the various tools are grouped based on the type of sRNA they are designed for. In the case of miRNAs, we focus only on the DL methods, since they are the current state-of-the-art in the field and the majority of the previous tools have already been described in numerous reviews.”
Some explanations are redundant. For example, sRNARFtarget is explained in l.395 and l.464, which duplicates the content. It would be better to change the structure of the paper and explain it in one place.
We thank the reviewere for this comment. The paragraph for sRNARFtarget has been kept in the relevant section “2.2.2.1. sRNA target interactions”; the duplicate has been deleted. The relevant entry in Table 1 has been moved under the “sRNA target prediction” label.
Sec 2.3: “Databases of sRNAs and sRNA:target interactions” Unlike the description of the tools in section 2.2, I think the description of the database is too limited. Since this manuscript is a review of the sRNA target prediction tools, it would be unnecessary to include this section.
The section "Databases of sRNAs and sRNA:target interactions” has been omitted.
l.138: “PR AUC” It is difficult to understand this explanation of PR AUC because how the curve is drawn is not explained.
The following line has been added to the Introduction: “The Precision-Recall curve is constructed by calculating and plotting the precision against the recall at a variety of thresholds."
l.502: “RNA-BP binding site” What does "BP" stand for? It is probably not a very common abbreviation.
“RNA-BP” has been changed to RNA-binding protein.